# Sliding friction over individual aromatic bonds correlates with bond order

Shinjae Nam[1,2,7], Lukas Hörmann [3,4,7], Oliver Gretz [1,7], Oliver T. Hofmann [5], Franz J. Giessibl [1,6] & Alfred J. Weymouth [1,6] ✉

Friction is ubiquitous, and has therefore been studied extensively to determine how it can be modified. Most experiments are not controlled down to the atomic level and encounter challenges with repeatability. We oscillate a tip ending in a single atom laterally over individual chemical bonds and measure the resulting energy dissipation. While one might expect the energy loss over aromatic bonds to be very similar, this is not the case. DFT-based simulations show that over aromatic bonds, the sliding friction correlates to bond order and is largely determined by the increased electron density between the atoms. Finally, we compare this to friction over hydrogen bonds and show that friction can be of the same magnitude but is due to interaction of the single atom asperity with the atoms of the hydrogen bond themselves. These findings show how friction can be tuned by adjusting the bond order of sliding surfaces.

Unveiling the dynamics and energy dissipation involved in atomic-scale motion is key to understanding surface catalysis[1–3], molecular motors[4,5], and single-molecule manipulation[6,7]. Despite significant progress in nanoscale friction[8–10] studies, due to challenges in atomistically defining the sliding surfaces, there are outstanding problems regarding reproducibility, isolating nonconservative interactions, and, in general, reconciling atomistic theory with experimental results[11]. This has prompted investigations with precise control at the single-atom scale[12–15]. We use a single atom asperity[16] as one sliding surface. High spatial resolution allows us to investigate individual chemical bonds and address the question of how the nature of a chemical bond affects sliding friction. Surprisingly, we find a large variety in sliding friction over different aromatic bonds. Density functional theory-based simulations yield excellent agreement with the data and reveal that sliding friction is correlated to bond order. Finally, we show that over hydrogen bonds, the maximum magnitude of sliding friction can be similar to friction over covalent bonds, that interaction is not with an increased electron density between the atoms. These findings offer new insights into atomic-scale motion and show that the frictional

properties of advanced materials[17] and nanodevices can be tuned by selecting the nature and order of chemical bonds at surfaces.

A full understanding of friction would allow not only the frictional responses of existing materials to be understood, but also provide the framework to design materials with desired frictional properties[17]. Modern theoretical descriptions of friction have shown that the main components to friction are small asperities, and that a full understanding requires considering interactions over a range of length scales down to the atomic level[18]. Friction is a challenging problem, even over surfaces that are approximately flat at the atomic scale. When an asperity slides over a surface, it often encounters fundamentally different kinds of chemical environments, including covalent bonds and hydrogen bonds. While macroscopic friction is the product of many asperities interacting with a sliding surface, to address fundamental questions, it is essential to understand friction down to the level of single atoms[12,14,19,20]. One might assume, for instance, that the energy loss when sliding over different aromatic bonds is the same. In this work, we show that even small changes in bond order are correlated with large differences in sliding friction. This is possible only by

[1]Faculty of Physics, University of Regensburg, Regensburg, Germany. [2]Center for Quantum Nanoscience, Institute for Basic Science (IBS), Seoul, South Korea. [3]Department of Chemistry, University of Warwick, Coventry CV4 7AL, UK. [4]Faculty of Physics, University of Vienna, Vienna 1090, Austria. [5]Institute of Solid State Physics, Graz University of Technology; NAWI Graz, Graz, Austria. [6]Regensburg Center for Ultrafast Nanoscopy (RUN), University of Regensburg, Regensburg, Germany. [7]These authors contributed equally: Shinjae Nam, Lukas Hörmann, Oliver Gretz. ✉e-mail: jay.weymouth@ur.de

directly measuring sliding friction using the smallest possible, best-defined asperity: A single atom.

## Results

### Measuring sliding friction over individual aromatic and hydrogen bonds

The experimental setup is shown in Fig. 1a. In frequency-modulation Lateral Force Microscopy (LFM), the tip oscillates laterally above the surface[21] (also see Methods and Supplementary Methods). The stiff qPlus sensor allows oscillation amplitudes smaller than interatomic distances, which are required for high spatial resolution. Before collecting data, the tip is functionalized with a single CO molecule[16], as sketched in Fig. 1b (also see Supplementary Methods). The CO-tip has the advantage of being chemically inert, which prevents changes to the tip and substrate (i.e., prevents wear of either sliding surface) during measurements. By characterizing the tip apex[22,23], we can perform reproducible data acquisition. The frequency-modulation technique allows conservative and non-conservative interactions to be independently measured via two feedback loops for resonance frequency and amplitude: The average energy dissipated per oscillation cycle, $E_{diss}$, and the frequency shift, $\Delta f$ (a measure of the conservative interaction), are simultaneously recorded. Spatially resolved energy dissipation data over covalent bonds are shown in Fig. 1c. We note that normal AFM experiments, with a vertically oscillating tip, cannot laterally slide over individual chemical bonds and can therefore not yield data as shown in Fig. 1c.

Figure 1d is a sketch of the mechanism of energy dissipation over a single chemical bond calculated using density functional theory-based (DFT-based) simulations. The simulations are explained later in more detail. As the metal apex moves left to right (i to ii), the CO deflects, and energy is stored as it would be in a torsional spring[22,24]. This deflection is also referred to as angle bending. At each position of the metal tip, the CO deflects to its local low-energy position, given by the sum of the energy stored in the spring and the interaction with the surface (described by the potential energy landscape of the CO with the surface). Energy can be stored in the torsional spring until the metal tip passes over the chemical bond and the CO snaps down (shown in the Supplementary Movie 1), exciting vibrations of the CO[25]. Note that the resulting vibrational excitations that transfer the energy loss into phonon modes[8,26,27] and electrical excitations[28,29] are not shown. A hysteresis loop opens[30] when the lateral forces, exerted on the tip, differ between forward and backward motion during one oscillation cycle. We are sensitive to the area enclosed by the closed path in the force-distance plot (the gray shaded region in the lateral force versus lateral position of the metal apex, shown in Fig. 1d), which is the energy dissipation $E_{diss}$.

The energy-dissipation signal probes the surface potential energy landscape via the O atom at the tip apex. In contrast to normal-force AFM measurements, where energy dissipation with a CO-tip is not observed over single chemical bonds, the measurement of dissipated energy during a lateral oscillation is inherently short-range because the only contributions to the measured signal are those that differ from the forward and backward paths within one oscillation cycle. As we showed previously, the signal decays with a decay length of 4 pm, which is much smaller than those reported for STM or normal-force AFM measurements[24]. It also means that the signal probes the potential energy landscape within a range of less than one Angstrom.

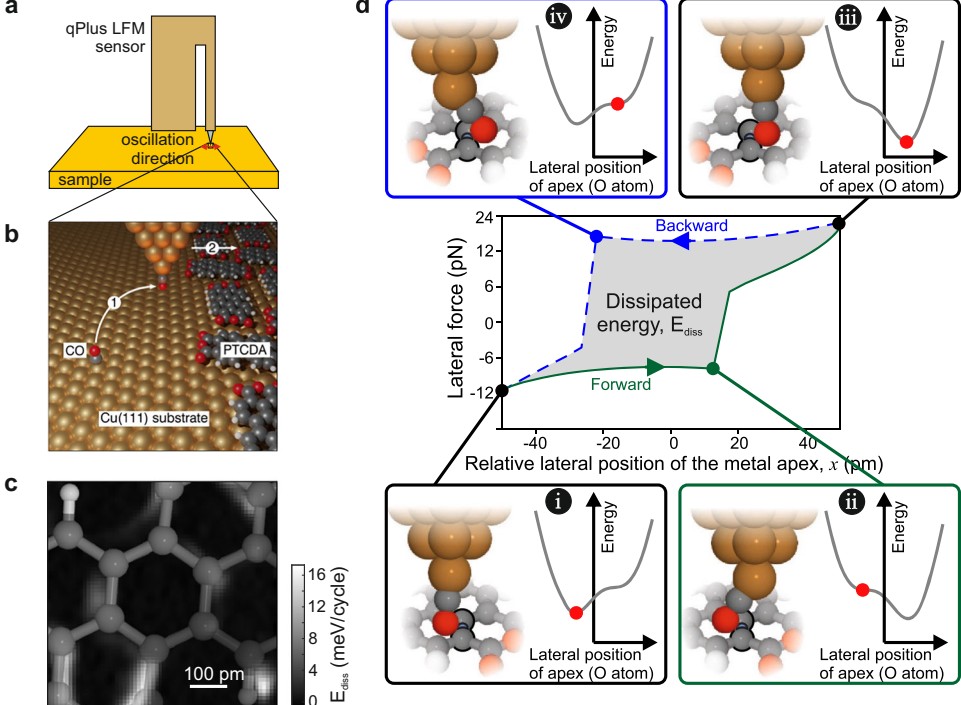

**Fig. 1 | Setup and mechanism of measuring sliding friction. a** Sketch of the experimental LFM (Lateral Force Microscopy) setup. A qPlus sensor is constructed so the tip oscillates laterally over the surface. **b** 1. A CO-tip is made and 2. data is acquired over an island of perylenetetracarboxylic dianhydride (PTCDA) molecules. Standard colours for spheres are used in this article: red indicates O, dark gray indicates C, white indicates H and copper indicates Cu. **c** Over single chemical bonds, energy loss can be measured. A map of $E_{diss}$ is shown with the half-transparent chemical structure of a PTCDA molecule, as determined by a simultaneously acquired LFM $\Delta f$ image (Supplementary Fig. 1). **d** The physical mechanism of $E_{diss}$: As the apex metal atom oscillates laterally over a chemical bond, the CO at the apex slides over and work is done. (The relative position x = 0 pm is the centre of the chemical bond.) At different positions of the apex metal atom (i, ii, iii, and iv), the potential energy landscape presented by the surface is different. Panels i, ii, iii, and iv show the total energy as a function of the deflection of the O atom at the tip apex (gray curve) and its corresponding actual lateral position (sketch on the left side and the red dot). Because the CO is flexible, it can be caught in a local energy minimum, as in ii and iv. The result is that the lateral forces exerted on the CO are different as the tip moves forward (i to ii) compared to when the tip moves backward (iii to iv).

To explore the sliding friction of various covalent and hydrogen bonds in a systematic way, we use perylenetetracarboxylic dianhydride (PTCDA) molecules adsorbed on Cu(111) (described more in Methods). This system provides a versatile platform for comparing friction over hydrogen bonds to friction over covalent bonds and for comparing sliding friction over covalent bonds of different bond orders.

## Energy dissipation is different over chemically similar bonds

Initially, we assumed that the interaction of the CO is predominantly with the two nearest carbon atoms, and that the energy dissipation as a function of height would have similar maximum values over all covalent bonds (assuming that they are oriented equivalently to the oscillation direction). To test whether sliding friction is indeed similar over various covalent bonds, we collected data over covalent (C-C) and hydrogen bonds (O···H) shown in Fig. 2a.

To determine the friction over individual chemical bonds, the maximum energy dissipation was evaluated and plotted as a function of the tip height as shown in Fig. 2b and c (as described in the Methods and Supplementary Methods). These covalent bonds were chosen because they are oriented in the same direction with respect to the direction of tip oscillation[31,32], as shown in Fig. 2a. Data over other bonds are shown in Supplementary Fig. 2. The heights of each curve (x-axis) were determined by the DFT-based simulation and represent the height of the unrelaxed O of the tip apex (300 pm closer than the metal tip apex atom) above the plane of the molecular adsorbates. Starting at a height where no measurable energy dissipation occurs and decreasing the tip-sample distance (tip height) over a chemical bond, an increase in $E_{diss}$ is observed. This is because the potential energy barrier that the surface presents to the apex becomes larger (as discussed later). Below a certain height, the CO can no longer snap during each oscillation cycle but is trapped on one side of the bond[24] (Supplementary Movie 1).

Because both sliding surfaces are controlled at the atomic level, we can reproducibly acquire data on friction in the contact regime. The high reproducibility of the data can be seen in the relatively small error bars, representing the standard deviation of $E_{diss}$ over various measurement sets.

Contrary to our initial hypothesis, the overall maximum energy dissipation values over various C-C covalent bonds differ notably, as shown by the spread in the maxima of the curves in Fig. 2b, which vary almost a factor of two from 13.1 to 23.6 meV/cycle. Moreover, the maximum energy dissipated is not always greater over covalent bonds than over hydrogen bonds, as shown in Fig. 2c: The energy dissipated over O···H(1) is 18.0 meV/cycle, which falls in the range of the observed $E_{diss}$ over covalent bonds.

To confirm the mechanism and understand the notable difference in the maximum $E_{diss}$ in more detail, we performed DFT-based simulations of $E_{diss}$, shown in Fig. 2d over the covalent bonds and in Fig. 2e over hydrogen bonds. These simulations include all interactions between the CO at the tip apex and the surface atoms, and are the gold standard for considering the interaction of the tip with the surface. (See Methods and Supplementary Methods.) The increase of $E_{diss}$ versus tip height for each curve is in excellent agreement with the data (Supplementary Fig. 3). The maximum values of $E_{diss}$ from the simulation are also in good agreement with those from the experiment. We note that in the experiment, the largest dissipation is first found over C-C(1), then C-C(4), C-C(2), and finally C-C(3), whereas in the simulation, this order is C-C(4), then C-C(3), C-C(2) and then C-C(1). We tentatively attribute this to the fact that the simulation, by necessity, considers a two-molecule supercell, whereas the experiment probes PTCDA lying in an incommensurate lattice. At the height where we measure the PTCDA dissipation, we do not see any impact from the copper substrate (discussed below); there are geometric differences in different adsorption sites (Supplementary Fig. 4). These distortions from the gas-phase planar geometry affect the potential energy

landscape and $E_{diss}$ (Supplementary Fig. 5). To verify the need for DFT-based calculations to determine the potential energy landscape, we also performed simulations using empirical atomic interactions[24,33]. These results, shown in Supplementary Fig. 6, show a poorer agreement with the experimentally-determined $E_{diss}$.

The DFT-based simulations show, in agreement with the data, that the maximum energy dissipation over hydrogen bonds can be greater or less than the maximum energy dissipation over covalent bonds. We note that the DFT-based simulations do not include variations in the experimental oscillation amplitude and thermal effects which we believe are responsible for the smooth decrease in energy dissipation at heights below the maximum. Future studies are needed to address these effects. Notably, large differences in energy dissipation are observed over chemically similar bonds, raising the question as to which mechanisms govern these differences.

## Energy dissipation over covalent bonds is explained by bond order

First, we consider what might lead to different maximum values of sliding friction across different aromatic bonds. One attribute of covalent bonds is their significant electron density between the two atomic cores. Ellner et al. showed that the CO tip does interact with the electron density of the bond itself[34]. Related work by Gross et al. demonstrated a correlation between the apparent length of a covalent bond in normal AFM images and its bond order[35].

To investigate this connection, we determined bond order from the DFT calculations of fifteen covalent bonds marked in Fig. 3a via the Mulliken Population Analysis[36] (described in the Supplementary Methods). We simulated the energy dissipation over each bond (assuming the tip oscillates perpendicularly over it) as a function of Mulliken bond order (Fig. 3b). The Mulliken bond order differs from the classically defined bond orders in chemistry: A single bond has a Mulliken bond order of 0.7 and a double bond has a Mulliken bond order of 1.4 (Supplementary Fig. 7). The aromatic bonds of benzene have a Mulliken bond order of 1.14. A linear fit of the energy dissipation versus the bond order yields a significant correlation of 0.7 with a confidence level of 99.8% that the correlation is not due to random chance.

We understand the correlation between bond order and energy dissipation as follows: According to Mulliken Population Analysis, a high bond order results from increased electron density between the bonding atoms. This elevated electron density increases Pauli repulsion, resulting in a more corrugated potential energy surface. Consequently, this leads to greater energy dissipation.

## Maximum energy dissipation over hydrogen bonds found at lower height

We then turn to understand the difference between sliding friction over covalent versus hydrogen bonds. $E_{diss}$ images were collected over an area with both covalent and hydrogen bonds in it (Fig. 4a, b). Then the $E_{diss}$ versus z curves were extracted (Fig. 4c, d). These curves are not the averaged curves as presented in Fig. 2, but single datasets. The relative z heights from the experimental maxima were extracted directly from the data. Then we simulated the $E_{diss}$ curves over these bonds (Fig. 4c, d). In both simulated and experimental data, the dissipation over hydrogen bonds is observed at lower tip heights (46 pm, as indicated by the distance between the vertical dashed lines in Fig. 4c, d).

While the atoms involved in the hydrogen bond are lower (Supplementary Fig. 8), the geometric height difference (< 25 pm) is not large enough to fully explain the experimentally observed and simulated height difference. Hydrogen bonds exhibit insignificant additional electron density between the bonding atoms (Supplementary Figs. 9 and 10). Therefore, over hydrogen bonds, the dominant interaction is not with an increased electron density between the atoms (covalent bond) but rather with the atoms themselves[34].

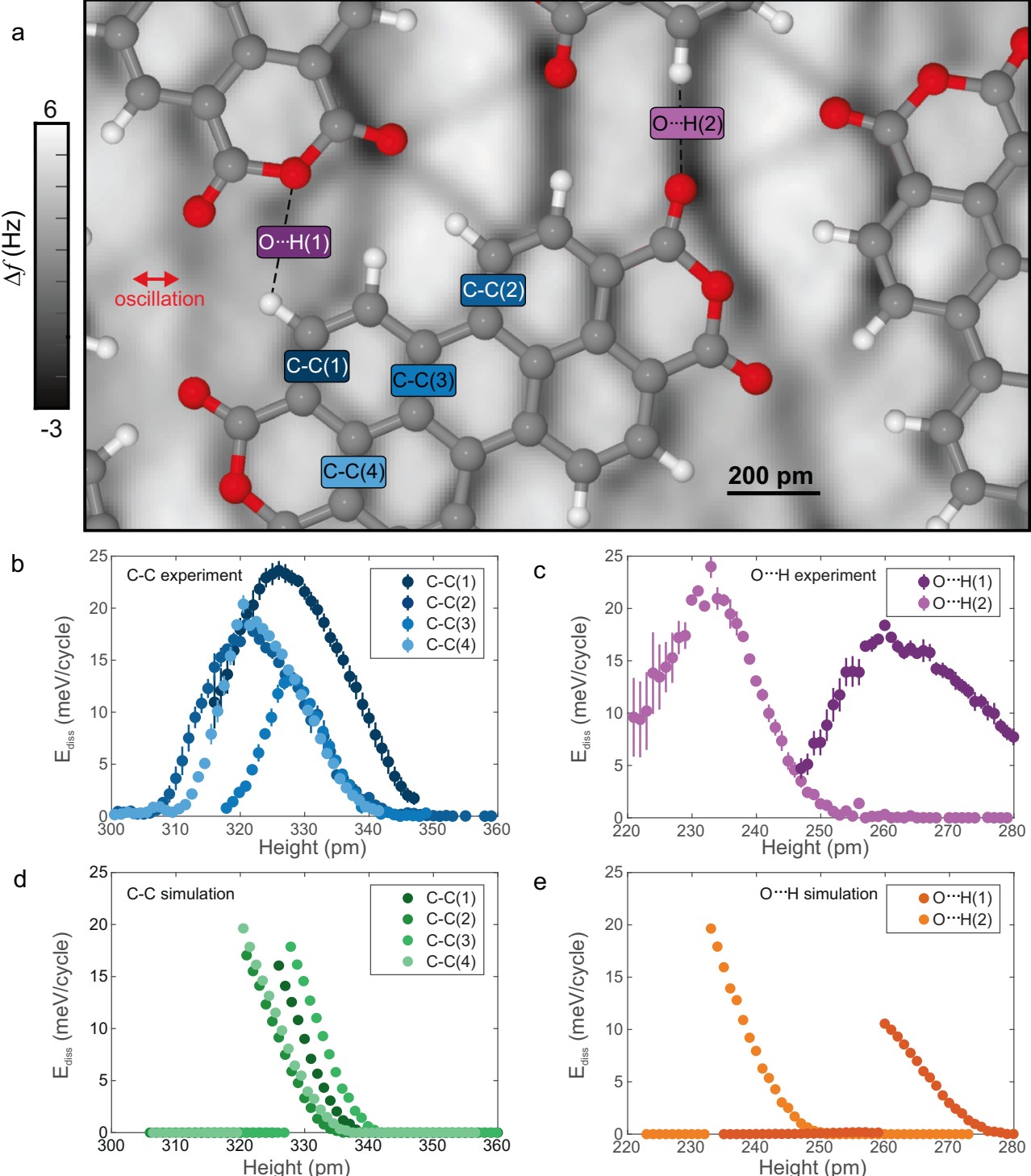

**Fig. 2 | Measuring sliding friction over covalent and hydrogen bonds. a** Area above the PTCDA island with several bonds identified. The background image is a LFM $\Delta f$ image, used to identify the position of the chemical bonds. Red indicates oxygen, gray indicates carbon, white indicates hydrogen. **b** $E_{diss}$ versus tip height over several covalent bonds. The data was acquired using the same CO tip and repeated 12 times. The maxima were shifted to align with the maxima from the simulation, shown in d. **c** $E_{diss}$ versus tip height over two hydrogen bonds. The data was acquired using the same CO tip as used for covalent bonds, and the maxima were set to align with the peaks in e. Data points in **b** and **c** are mean values of 12 technical replicates. The error bars in **b** and **c** represent the standard deviation. **d** DFT-based simulation output of $E_{diss}$ over covalent (C-C) and **e** hydrogen (O⋯H) bonds. Source data are provided in the Source Data file.

The relationship that we observed between bond order and sliding friction over aromatic bonds does not hold for hydrogen bonds: The Mulliken bond order of hydrogen bonds is small (Supplementary Fig. 10) and yet the magnitude of sliding friction is similar to that over aromatic bonds (Fig. 2). The reason that the dissipation is similar in magnitude is because the energy barrier presented laterally over hydrogen and covalent bonds are themselves of a similar shape, as shown in Fig. 4e, f.

We were unable to find a similar correlation between friction and bond character for the OH bonds. In Supplementary Fig. 11 we show

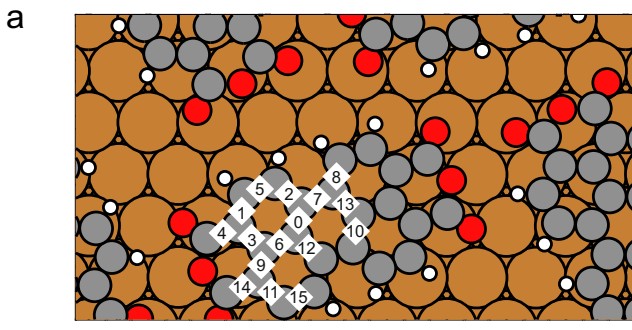

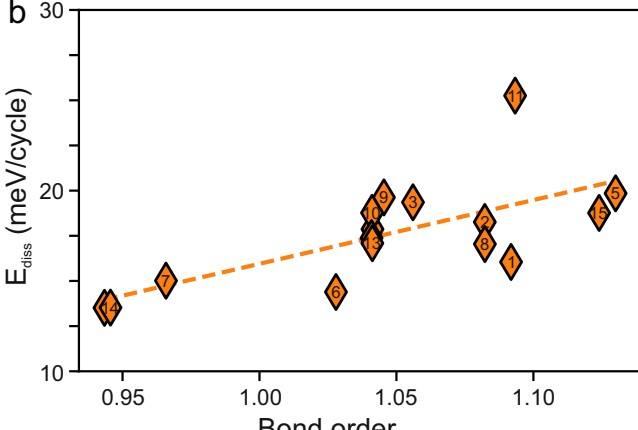

**Fig. 3 | Sliding friction over covalent bonds as a function of bond order. a** Map of the covalent bonds considered. Red indicates oxygen, gray indicates carbon, white indicates hydrogen, and copper indicates copper. **b** $E_{diss}$ as a function of bond order. Within these calculations, a single bond has a Mulliken bond order of 0.7, whereas a double bond has a Mulliken bond order of 1.4 (Supplementary Information). The linear fit (dashed line) is defined by the coefficient 35.3 meV/cycle and the intercept −19.3 meV/cycle. Source data are provided in the Source Data file.

that the maximum energy dissipated over a OH bond is not a monotonic function of the distance between atoms.

## Discussion

Sliding friction is a non-conservative force that acts when two objects slide against each other and opposes the relative motion between the two surfaces. Non-conservative forces are those yield non-zero work when an object is moved in a closed path. During each oscillation, the metal apex is slid forward and backward over the surface in a closed path, and if work is done on the sensor it is recorded as the energy dissipation.

We made use of the phenomenon of molecular snapping to investigate friction between two sliding surfaces that are characterized at the single-atom level. Both experimental data and DFT-based simulations revealed that sliding friction over covalent bonds can vary strongly, and it is not the case that the maximum friction over covalent bonds is greater than the maximum friction over hydrogen bonds. DFT-based simulations demonstrate that the wide variety in sliding friction over aromatic bonds can be understood in that sliding friction is correlated to bond order.

While the potential energy landscape determines friction at individual chemical bonds, its origin differs between covalent and hydrogen bonds. In covalent bonds, the potential energy landscape is dominated by increased electron density between atoms, leading to a correlation between bond order and sliding friction. Over hydrogen

bonds, the potential energy landscape is dominated by interaction with the atoms themselves.

These observations not only enhance our understanding of friction from the atomic scale, but also help guide the realizations of atomically precise materials with specific friction characteristics. For example, target friction values[17] can be engineered[37,38] by selecting chemical bonds of an appropriate order.

## Methods

### Mapping sliding friction over single chemical bonds

Data were collected on a low-temperature (He-bath) STM/AFM system manufactured by CreaTec Fischer & Co. GmbH that had been modified for LFM sensors, as shown in Fig. 1a and described in refs. 13,39. The oscillation direction is defined by the sensor design and cannot be changed during operation. The control electronics were Nanonis electronics manufactured by SPECS Zurich GmbH. Data were collected with the Nanonis SPM Control Software version Generic 5 and analysed with MATLAB 2024.

The centre frequency of the sensor was 41.522 kHz, and the quality factor was $Q = 41911$. The LFM sensor was a 0.8 length qPlus sensor[40]. The stiffness $k = 2144$ N m⁻¹ was calculated by taking into account the length of the tip[41]. The stiffness of the CO at the apex was previously experimentally determined[22]. The apex did not change during measurements, as shown in the Supplementary Methods (Supplementary Fig. 12). The amplitude was calibrated by imaging an adsorbate with STM with and without tip oscillation and fitting the image with oscillation to a theoretically calculated image using the data without oscillation[39]. The excitation data shown in Fig. 2D were collected with an amplitude of oscillation of 50 pm. In our previous work, we investigated the influence of other amplitudes and showed that the snapping was the dominant contribution as long as the amplitude is small enough to only snap over one bond[24]. Excitation at these heights was not influenced directly by the underlying Cu substrate, as can be seen in Supplementary Fig. 13. We also investigated the influence of the Cu substrate theoretically, as can be seen in Supplementary Fig. 6.

In frequency-modulation mode, the amplitude of the oscillation is set and maintained by a feedback loop[42]. Energy dissipation due to surface interaction requires an increase in the drive to maintain the oscillation. The drive signal is recorded and converted into a value that describes the energy loss per cycle[43].

Cu(111) was prepared using standard sputter-and-anneal cycles. PTCDA (perylenetetracarboxylic dianhydride) was evaporated from a home-built evaporator. The evaporator had a stainless-steel exterior and molecules were placed in a glass cell with a tungsten wire wrapped around them. CO was deposited from the gas phase via a precision leak valve. Coverage for PTCDA and CO was determined via STM images. Images of the area were then collected and linescans were taken between the atomic centres, as shown in Supplementary Fig. 14.

### DFT modelling

To find a mechanistic explanation for the observed differences in energy dissipation, the sliding surfaces were simulated. DFT modelling was used to determine the atomic positions of two PTCDA molecules in a supercell on the Cu(111) surface. While the experimentally-observed superstructure appears to be incommensurate, a commensurate structure was used as has been previously discussed in the literature[44] as a necessary balance between accuracy and computational effort.

The interaction of the tip with the surface is predominantly via the CO at the apex[45]. To determine the potential energy landscape, therefore, requires calculating the interaction of the CO at each lateral and vertical position above each chemical bond with a theoretical model that explicitly includes quantum mechanical interactions. DFT yields the necessary accuracy, but the immense number of energy and force evaluations required to determine an accurate potential energy

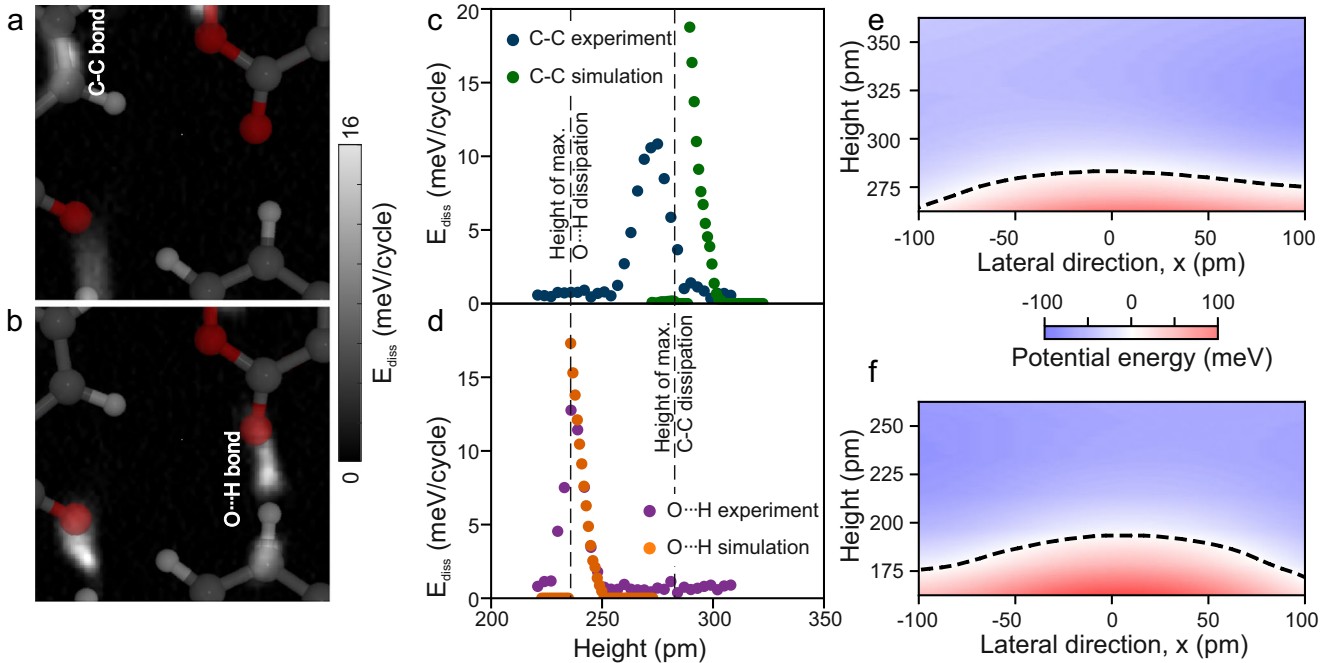

**Fig. 4 | Sliding friction over covalent versus hydrogen bonds. a** $E_{diss}$ image taken over several covalent and hydrogen bonds shows contrast above the covalent bond. **b** At a height 46 pm lower, friction over the hydrogen bond is more significant. **c, d** Experimental and calculated $E_{diss}(z)$ curves of the **c** covalent and **d** hydrogen bonds. **e** Potential energy of the O-atom of the tip CO above the C-C bond. The dashed line indicates zero energy in (**e** and **f**). **f** Potential energy landscape above the hydrogen bond. Source data are provided in the Source Data file.

landscape renders the sole use of DFT intractable. We overcome this hurdle by using a machine-learning model based on DFT training data to generate an accurate potential energy landscape. Because the interaction is predominantly via the oxygen atom[22,46], we performed this calculation for a vertically oriented CO molecule. The training data for this machine-learning model was also calculated with DFT (see Supplementary Methods).

This potential energy landscape was then one input into a snapping model (described in more detail in the Supplementary Methods), which simulates the oscillation of the tip at a certain point, assuming that the CO at the apex responds to applied forces as a torsional spring and that the atoms of the surface are fixed[24,33]. At each point of the oscillation, described by the position of the metal apex atom to which the CO is bound, the CO is allowed to relax. This relaxation is governed by its interaction with the surface, as previously described, and by the deflection of the CO, which introduces an additional force modeled as a torsional spring. Then, the lateral force component of the surface acting on the tip in the direction of the tip oscillation is recorded. These assumptions are similar to those of the successful Probe Particle Model (PPM)[33]. The PPM is used to simulate $\Delta f$ images of normal-force AFM images with a functionalized tip[33]. Our model extends the PPM by forcing the CO to remain in a local energy minimum, thereby allowing energy dissipation.

The calculated lateral forces over one oscillation cycle over a covalent bond are shown in Fig. 1d. The hysteresis between forward and backward motion supports our previously proposed hypothesis that the energy dissipation is due to mechanical deformation of the tip apex as the CO is cocked and snaps over a single bond. The energy dissipation is the area within the hysteresis loop.

We investigated the angular dependence of the energy dissipation in Supplementary Fig. 15. The low-energy vibrational modes of the PTCDA adlayer were also investigated (Supplementary Fig. 16 and Supplementary Table 1). The corresponding stiffnesses of these modes are approximately 100 times higher than that of the CO, indicating that relaxation of the molecule can be ignored during the snapping.

## Data availability

The raw experimental data used in this study are available on the University of Regensburg's Publication Server under the DOI 10.5283/epub.58384 [https://epub.uni-regensburg.de/58384/]. The DFT results used in this study are available on the NOMAD database under the DOI 10.17172/NOMAD/2024.06.02-1 [https://nomad-lab.eu/prod/v1/gui/dataset/id/BmacZB6JRwG3zLuygiaMAA]. The $E_{diss}$ data generated in this study are provided in the Source Data file. Source data are provided with this paper.

## Code availability

Code for the model (Python, version 3) is available at https://doi.org/10.5281/zenodo.14923664.

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

## Acknowledgements

Funding from the German Research Foundation (Deutsche Forschungsgemeinschaft – DFG – Project number 397771090 and Project number 444750204); from the Steiermärkische Landesregierung Forschungsförderungsprogramm „Unkonventionelle Forschung"; and from the UKRI Horizon Guarantee EP/Y024923/1 is gratefully acknowledged. Funding is also gratefully acknowledged from the DFG through GRK 2905 (project-ID 502572516). This research was also funded in part by the Austrian Science Fund (FWF) [10.55776/Y1157]. Computational results have been obtained using the Vienna Scientific Cluster (VSC) and ARCHER2, a high-performance computing service provided by the UK's national supercomputing facility. A.J.W. would like to thank Marco Weiss and Sophia Schweiss for comments on the manuscript.

## Author contributions

O.G. and A.J.W. conceptualized the investigation. The methodology was developed by S.N., O.G., L.H., A.J.W., F.J.G. and O.T.H. The investigation was conducted by O.G., S.N., L.H. and A.J.W. Visualization was carried out by S.N., L.H. and A.J.W. Funding acquisition was performed by A.J.W., L.H., O.T.H. and F.J.G. The project administration was handled by A.J.W. Supervision was provided by A.J.W., O.T.H. and F.J.G. The original draft was written by A.J.W., L.H. and S.N. Review and editing were performed by all authors.

## Funding

## Competing interests

FJG holds patents for the qPlus sensor. Patent applicant and inventor F. J. Giessibl holds US patent 8393009 (since 5 March 2013), which was used to acquire the experimental data. The remaining authors declare no competing interests.
