## [Transparent Peer Review file · Nature Communications]

Sliding friction over individual aromatic bonds correlates with bond order

Corresponding Author: Dr Alfred Weymouth

Version 0:

Reviewer comments:

Reviewer #1

(Remarks to the Author)

The manuscript titled “Sliding Friction Over Individual Covalent Bonds Correlates with Bond Order” by Shinjae Nam et al. presents an atomic-scale study of sliding friction using a carbon monoxide tip to achieve a single-atom asperity, providing evidence that the energy dissipation of an oscillating force sensor is influenced by the chemical bonds of molecules. The combination of experiment and DFT-based simulations offers insights into molecular sliding friction at the atomic scale. The authors’ concept appears novel, especially using frequency-modulation Lateral Force Microscopy. However, we, as part of the co-review program, believe that the experimental data provided is inadequate to support their arguments. Our main concern relates to the mechanism of dissipated energy. The authors utilized a very soft CO tip as a friction force sensor. Consequently, all signals appear to emanate from the tip, rather than the molecule adsorbed on the surface. Additionally, the mechanism of dissipation described in the manuscript lacks clarity. The authors stated that differences in the origin of force induce varying heights in the dissipation energy peaks (such as hydrogen bonds and C-C bonds). However, it seems the peak position is merely related to the strength of lateral force. The bond order analysis is quite narrow, making it challenging to assess the validity of the analysis. Thus, we cannot support this manuscript for publication in Nature Communications in the current form. Below, we summarize our comments:

1. Figure 1d

Figure 1d is quite difficult to understand. During the measurement, the tip should oscillate parallel to the surface. We believe the schematic illustrates a snapshot from one oscillation cycle, but there is no clear explanation. We found it unclear how the authors obtained the force loop from DFT calculations. The tip softness is a critical factor in friction measurement, yet we found no investigation of this in the manuscript. Furthermore, the authors used a single oscillation amplitude (50 pm), though, similar to the friction loop, oscillation amplitude is another key parameter. We strongly recommend a thorough investigation into these points.

2. line 5 page 5

The authors described “To test this theory”. Which theory?

3. line 10 page 5

“These covalent bonds were chosen because they are oriented in the same direction with respect to the direction of the tip oscillation”.

Our concern relates to the oscillation direction which is not aligned perpendicular to the bond orientation. This misalignment could significantly impact the measurement accuracy and the interpretation of the friction phenomena.

4. line 15 page 5

“This is because the potential.....”

The authors should provide an explanation for why the peak is Gaussian-shaped.

5. Figure 2b and 2d

The dissipation maxima for different bonds in Figure 2b do not agree with the simulations presented in Figure 2d. In Figure 2b, the dissipation maxima follow the order C-C(1) > C-C(4) > C-C(2) > C-C(3), whereas in the simulation shown in Figure 3, the order is C-C(1) > C-C(2) > C-C(4) > C-C(3). The authors should provide an explanation for this discrepancy.

6. line 16 page 7

"We note that the DFT-based simulations do not include variations in the experimental oscillation amplitude and thermal effects.."

We understand that incorporating these parameters is challenging. The authors claim these effects lead to a smooth decrease in energy dissipation. We agree, yet if this is the case, thermal effects (and oscillation) should lower the peak height. However, the authors achieved excellent agreement between experiment and theory regarding the maximum values of E_{diss} , which seems odd.

7. the last line on page 8.

"Consequently, this leads to greater energy dissipation."

This is too simple. The authors should provide more explanation.

8. Figure 3

The conclusion that energy dissipation correlates with bond order comes from Figure 3, which is based on simulations, not experimental data. As mentioned in comment 5, the relative magnitudes of the dissipation maxima obtained in experiments do not align with those from the simulations. So, the conclusion is not convincing. The experimental dissipation data corresponding to all C–C bonds should be provided. It is understandable that the orientations of different C–C bonds may vary, and molecule manipulation could help address this issue. Alternatively, studying a molecule with single, double, and triple bonds in the same structure would offer more convincing support.

9. It is recommended to also consider the C–O bonds within the molecule to determine if they follow the proposed correlation between bond order and energy dissipation.

10. Figure 4ef

It is difficult to understand the figures.

11. The discussion on dissipation over hydrogen bonds is insufficient. Why is the energy dissipation over some hydrogen bonds larger than that over covalent bonds? The order is a key factor for covalent bonds, what determines dissipation in the case of hydrogen bonds? Why different hydrogen bonds exhibit varying maximum dissipation values?

12. For such high-sensitivity measurements, has the author considered the effects of molecular vibrations. This point should be discussed.

Minor

1. Line 10 on page 3

"energy dissipated per cycle" of what?

2. Line 11 on page 3

"The heights of each curve (x-axis) were determined by the DFT..."

We suppose that the origin of the heights was determined by the DFT... or something missing?

(Remarks on code availability)

Reviewer #2

(Remarks to the Author)

In this paper the authors propose that there is a correlation between the energy dissipation measured in frequency-modulation LFM experiments and the amount of electronic density in the region of "covalent" bonds estimated with Mulliken analysis from DFT calculations of the PTCDA adsorbed molecule. However, the correlation shown in Fig. 3, which should constitute the main result, is evaluated taking into consideration the simulated dissipation energy values, rather than the experimental ones. Indeed, the order of the E_{diss} maxima (c1, c4, c2, c3) resulting from the experiment, shown in Fig. 2b, is not the same as that (c1, c2, c4, c3) of the computational ones, shown in Fig. 2d. Moreover, in this latter case c4 and c2 are clearly different, while experimentally they are almost identical.

The computational E_{diss} strongly depends on the constraints imposed in the model, which in the present case are substantial, being frozen all the degrees of freedom of the tip, PTCDA, and substrate. With the CO the only unconstrained object in the model, it is hard to believe that the calculated E_{diss} can quantitatively resemble the experimental values. The failure to fully reproduce the experimental trend implies that important contributions to the dissipation are missing in the model. As a matter of fact the removal of almost all dynamical effects, with the force always calculated for the relaxed positions, appears to be questionable.

The correlation between dissipation and bond order is limited to four different C-C bonds, which makes it rather uncertain the generalization to any "covalent" bond as suggested by the manuscript title. Indeed, the fact that almost the same amount of dissipation is observed for the case of hydrogen bonds makes the connection to bond-order even less general and more specific to C-C bonds in the PTCDA molecule.

In summary, although the paper presents very interesting experimental results their interpretation in relation to "sliding" friction is not convincing. The modeling of the process is not well explained even in the supplementary information. There are still unanswered open issues, such as the dissipation from the hydrogen bonds and the gaussian shape of the

experimental Ediss as a function of the separation, and too many assumptions in the model and its interpretation, whose effects on the estimate of the dissipation remain unclear.

Minor comments:

- Several important details of the studies have been confined to the Supplementary information, whose redaction has not been careful enough. Figures S2 show color coded surfaces, without any indication of which property is represented by the colors neither a colormap is shown to associate them to any actual value. Figure S2 is wrongly referenced several times (— > S3 ?)
- Definition of the color coding is also missing for figure 4e and 4f.

(Remarks on code availability)

Reviewer #3

(Remarks to the Author)

(Remarks on code availability)

Reviewer #4

(Remarks to the Author)

(Remarks on code availability)

Version 1:

Reviewer comments:

Reviewer #1

(Remarks to the Author)

We (co-review with an early career researcher) appreciate the efforts that the authors have made to address our (concerns and those of both reviewers). We find that many of the issues have been resolved, and the manuscript has been improved.

However, our main concern (comment 5) and Reviewer 2's main concern remain unaddressed. This point is central to the manuscript's main finding, as reflected in the title: "Sliding friction over individual aromatic bonds correlates with bond order." Although the analysis of the experimental data relies heavily on theoretical calculations, there is an inconsistency between experiment and theory. In the revised manuscript, the authors attribute this inconsistency to commensurate (theory) vs. incommensurate (experiment) lattices. While we understand that calculations for incommensurate lattices are nearly impossible, the underlying mechanism has not been clearly explained. The authors did not provide a clear rationale for why minor changes in bond length are caused by adsorption.

The authors also explained the mechanism of dissipation (R1.3). If this scenario is correct, one would expect to observe small lateral shifts in the peak position between forward and backward scans. The proposed mechanism also relies heavily on theoretical calculations, as there is no experimental data supporting this idea.

The dissipation signal between the molecules, at the hydrogen bonding sites, remains unclear. The authors propose some complex possibilities, but is it possible that the molecule is simply being laterally moved by the tip's oscillation?

Regarding R1.15, "Therefore we assume that the effects of thermal fluctuation at 5.8 K are not very large." I fully agree with this statement, but in the main text the authors write, "We note that the DFT-based simulations do not include variations in the experimental oscillation amplitude and thermal effects which we believe are responsible for the smooth decrease in energy dissipation at heights below the maximum." We think the phrase "we believe are responsible for the smooth decrease in energy dissipation at heights below the maximum" conflicts with the authors' previous argument.

We fully agree with Reviewer 2's comment: "Their interpretation in relation to 'sliding friction' is not convincing." The definition of "sliding friction" is unclear, even though the authors replied to our comments (R1.8). Thus, the current title might not be appropriate.

Minor:

The term "torsional spring" sounds odd, as there is no torsional movement of either the tip or the molecule.

(Remarks on code availability)

Reviewer #4

(Remarks to the Author)

(Remarks on code availability)

Reviewer #5

(Remarks to the Author)

The modifications made by authors in the revised manuscript have addressed the concerns made by reviewer#2 and improved the manuscript in a way that it can be deemed sufficient.

(Remarks on code availability)

Version 2:

Reviewer comments:

Reviewer #1

(Remarks to the Author)

We (co-review with an early career researcher) appreciate the efforts that the authors have made to address our concerns and to revise their manuscript, particularly Supplementation Table 1 and Fig. 17. We do not have any further questions regarding the manuscript.

(Remarks on code availability)

I am not expert on python.

REVIEWER COMMENTS

Reviewer #1 (Remarks to the Author):

The manuscript titled “Sliding friction over individual covalent bonds correlates with bond order” by Shinjae Nam et al. presents an atomic-scale study of sliding friction using a carbon monoxide tip to achieve a single-atom asperity, providing evidence that the energy dissipation of an oscillating force sensor is influenced by the chemical bonds of molecules. The combination of experiment and DFT-based simulations offers insights into molecular sliding friction at the atomic scale. The authors’ concept appears novel, especially using frequency-modulation Lateral Force Microscopy. However, we, as part of the co-review program, believe that the experimental data provided is inadequate to support their arguments.

(R1.1) We thank the reviewer for recognizing that this manuscript offers novel insights into friction at the atomic scale. In order to be more precise about the key finding of the article, we have taken their overall advice and changed the title of the paper to

“Sliding friction over individual aromatic bonds correlates with bond order”

Furthermore, we now provide more experimental data of sliding over covalent bonds in Supplementary Fig. 7, as discussed in more detail in our reply R1.17.

Our main concern relates to the mechanism of dissipated energy. The authors utilized a very soft CO tip as a friction force sensor. Consequently, all signals appear to emanate from the tip, rather than the molecule adsorbed on the surface.

(R1.2) The tip is a probe that is used to measure the surface. As the properties of the tip (apex termination) remain constant for the duration of the measurements, then the differences as we scan over the surface are a measurement of the surface. To reinforce that the tip has not changed during measurements, we now show characterization data before and after measurements in the Supplementary Methods (Supplementary Fig. 2).

Furthermore, we have now made the interaction mechanism clearer – that this is a probe of the potential energy landscape presented by the surface. This is now addressed on page 4 of the main manuscript with new additional text:

“The energy dissipation signal probes the potential energy landscape of the surface via the O-atom at the apex of the tip. In contrast to normal-force AFM measurements, where energy dissipation with a CO-tip is not observed over single chemical bonds, the measurement of dissipated energy during a lateral oscillation is inherently short-range because the only contributions to the measured signal are those that differ from the forward and backward paths within one oscillation cycle. As we showed previously, the signal decays with a decay length of 4 pm, which is much smaller than those reported for STM or normal-force AFM measurements²⁴. It also means that the signal probes the potential energy landscape within a range of less than one Angstrom.”

Additionally, the mechanism of dissipation described in the manuscript lacks clarity.

(R1.3) While the time-resolved dynamics of the dissipation are not the focus of this manuscript, we propose that the CO snaps down, exciting a vibration, and that this local vibrational excitation is converted into bulk phonons in the tip. This mechanism has been suggested in our earlier publications. To clarify this better, we changed the text on page 4:

“As the metal apex moves left to right (*i* to *ii*), the CO deflects, and energy is stored as it would be in a torsional spring^{22,24}. At each position of the metal tip, the CO deflects to its local low energy position, given by the sum of the energy stored in the spring and the interaction with the surface (described by the potential energy landscape of the CO with the surface). Energy can be stored in the torsional spring until the metal tip passes over the chemical bond and the CO snaps down (shown in the Supplementary Movie), exciting vibrations of the CO²⁵. Note that the resulting vibrational excitations that transfer the energy loss into phonon modes^{8,26,27} and electrical excitations^{28,29} are not shown. A hysteresis loop opens³⁰ when the lateral forces, exerted on the tip differ between forward and backward motion during one oscillation cycle. We are sensitive to the area enclosed by the closed path in the force-distance plot (the gray shaded region in the lateral force versus lateral position of the metal apex, shown in Fig. 1d), which is the energy dissipation E_{diss} .”

We also included a citation to Hörmann and Mauer (JACS Au, doi:10.1021/jacsau.5c00931 now Ref. 25). In this paper, the dissipation process is explained in more detail in the time domain.

The authors stated that differences in the origin of force induce varying heights in the dissipation energy peaks (such as hydrogen bonds and C-C bonds). However, it seems the peak position is merely related to the strength of lateral force.

(R1.4) The measured energy dissipation signal is a result of the CO at the tip apex being slid laterally over a potential energy barrier. As we describe in R1.3 above, the energy dissipation is an integral of the lateral force over an oscillation cycle of the tip. As the amplitude of oscillation is the same for all data reported in this manuscript, the reviewer is correct to observe that different values of dissipated energy are related to lateral force. However, the apex of the tip can follow a complex path in three dimensions during the oscillation and the energy dissipation is a measure of the difference between lateral force forward and backward. Therefore we feel it is safer to remain with the discussion of energy dissipation instead of discussing purely lateral forces.

The interaction of the CO at the apex is different over a hydrogen bond versus over a covalent bond. In order to strengthen this claim, we have added Supplementary Fig. 4, which we reference in the main text. In it, we show the important electron density which the CO is sensitive to. The increased electron density when sliding over atoms with covalent bonds is much higher than over the hydrogen bonds. While our original submission included Supplementary Fig. 12 (new figure name), we believe Supplementary Fig. 4 is clearer. We also discuss this more explicitly in the main text (p. 8):

“Hydrogen bonds exhibit insignificant additional electron density between the bonding atoms (Supplementary Figs. 4 and 12). Therefore over hydrogen bonds, the dominant interaction is not with an increased electron density between the atoms (covalent bond) but rather with the atoms themselves³⁴.”

The relationship that we observed between bond order and sliding friction over aromatic bonds does not hold for hydrogen bonds: The Milliken bond order of hydrogen bonds is small (Supplementary Fig. 12) and yet the magnitude of sliding friction is similar to that over aromatic bonds (Fig. 2). The reason that the dissipation is similar in magnitude is because the energy barrier presented laterally over hydrogen and covalent bonds are themselves of a similar shape, as shown in Figs. 4 e and f.”

And in the Summary and Outlook:

“While the potential energy landscape defines friction over individual chemical bonds, its origin is different over covalent versus hydrogen bonds. Over covalent bonds the potential energy landscape is dominated by the increased electron density between the atoms, resulting in the correlation between bond order and sliding friction. Over hydrogen bonds, the potential energy landscape is dominated by interaction with the atoms themselves.”

The bond order analysis is quite narrow, making it challenging to assess the validity of the analysis.

(R1.5) We understand the comment. Our observations – which surprised us – was how large the variety of sliding friction was over these seemingly-similar bonds. This prompted us to study them in more detail which led us to propose that the sliding friction correlates with bond order. Based on this reviewer’s careful feedback, as we mentioned in R1.1, we have changed the title to better reflect the important result.

Thus, we cannot support this manuscript for publication in Nature Communications in the current form. Below, we summarize our comments:

(R1.6) Again, thank you very much for your time and effort carefully critiquing our manuscript. We hope that after considering the significant changes we have made, you will recommend it for publication.

1. Figure 1d

Figure 1d is quite difficult to understand. During the measurement, the tip should oscillate parallel to the surface. We believe the schematic illustrates a snapshot from one oscillation cycle, but there is no clear explanation.

(R1.7) To address this, we have changed the figure caption for 1d:

“The physical mechanism of E_{diss} : As the apex metal atom oscillates laterally over a chemical bond, the CO at the apex slides over and work is done. (The relative position $x=0$ pm is the centre of the chemical bond.) At different positions of the apex metal atom (*i*, *ii*, *iii* and *iv*), the potential energy landscape presented by the surface is different. Panels *i*, *ii*, *iii* and *iv* show the total energy as a function of the deflection of the O atom at the tip apex (gray curve) and its corresponding actual lateral position (sketch on the left side and the red dot). Because the CO is flexible, it can be caught in a local energy minimum, as in *ii* and *iv*. The result is that the lateral forces exerted on the CO are different as the tip moves forward (*i* to *ii*) compared to when the tip moves backwards (*iii* to *iv*).”

We have also included in the main text a reference to the Supporting Movie, which we created to clarify the mechanism:

“Energy can be stored in the torsional spring until the metal tip passes over the chemical bond and the CO snaps down (shown in the Supporting Movie), exciting vibrations of the CO²⁵.”

We found it unclear how the authors obtained the force loop from DFT calculations.

(R1.8) Based on this comment, we have re-structured the Methods: Simulation section in the Supporting Information (which is referenced in the above-mentioned Methods section) to make the simulation and modelling clearer. In essence there are three components to simulating energy dissipation:

1. Relax the PTCDA on the surface (DFT-based determination of the adlayer structure)

2. Describe the potential energy surface that the CO encounters above the surface using a machine-learning algorithm trained on DFT data (Interaction of the CO molecule with the surface)
3. Simulate an oscillation of the tip with the above potential energy landscape and considering the tip as a torsional spring (Calculating dissipation)

The energy dissipation is then calculated with the equation (provided in the Supporting Information):

$$E_{diss} = 2\pi A f_0 \int_0^{1/f_0} F \sin(2\pi f_0 t) dt$$

Within the DFT modelling section at the end of the manuscript, we have added two sentences:

“At each point of the oscillation, described by the position of the metal apex atom to which the CO is bound, the CO is allowed to relax. This relaxation is governed by its interaction with the surface, as previously described, and by the deflection of the CO, which introduces an additional force modelled as a torsion spring. Then the lateral force component of the surface acting on the tip in the direction of the tip oscillation is recorded.”

The tip softness is a critical factor in friction measurement, yet we found no investigation of this in the manuscript.

(R1.9) We agree with the reviewer that this is very important. This has been the subject of a previous investigation of ours. Within the Methods section, we now include the sentence:

“The stiffness of the CO at the apex was previously experimentally determined²².”

Furthermore, the authors used a single oscillation amplitude (50 pm), though, similar to the friction loop, oscillation amplitude is another key parameter. We strongly recommend a thorough investigation into these points.

(R1.10) We agree with the reviewer about the importance of amplitude. This was a major component of a previous investigation. We now include in the Methods section:

“In our previous work, we investigated the influence of other amplitudes and showed that the snapping was the dominant contribution as long as the amplitude is small enough to only snap over one bond²⁴.”

2. line 5 page 5

The authors described “To test this theory”. Which theory?

(R1.11) We have since rephrased this paragraph:

“Initially, we assumed that the interaction of the CO is predominantly with the two nearest carbon atoms, and that the energy dissipation as a function of height would have similar maximum values over all covalent bonds (assuming that they are oriented equivalently to the oscillation direction). To test whether sliding friction is indeed similar over various covalent bonds, we collected data over covalent (C-C) and hydrogen bonds (O···H) shown in Fig. 2a.”

3. line 10 page 5

“These covalent bonds were chosen because they are oriented in the same direction with respect to the direction of the tip oscillation”.

Our concern relates to the oscillation direction which is not aligned perpendicular to the bond orientation. This misalignment could significantly impact the measurement accuracy and the interpretation of the friction phenomena.

(R1.12) To observe the variation above the bonds it is not necessary that they are perpendicular to the oscillation direction, rather that we can compare bonds that have the same angle with the oscillation direction.

We understand that the dissipation as a function of angle to the bond is an important parameter and now include a theoretical calculation showing the dependence of the dissipation as a function of the angle over an exemplary C-C and OH bond in the supporting material:

Dependence of oscillation direction on dissipation

Supplementary Figure 9 | Energy dissipation is plotted as a function of the oscillation direction. An angle of 0° means that the oscillation direction is perfectly perpendicular to the bond. (A) Data for tips of various stiffness (torsional stiffness of the CO) over a covalent bond and (B) over a hydrogen bond. For angles $\pm 10^\circ$, the dissipation does not strongly change.

4. line 15 page 5

“This is because the potential....”

The authors should provide an explanation for why the peak is Gaussian-shaped.

(R1.13) The sentence that the reviewer has quoted reads, “This is because the potential energy barrier that the surface presents to the apex becomes larger”. To illustrate this better, we have changed Figs. 4 e and f so that the lateral energy barrier in the direction of the tip oscillation is clearer. As the metal apex lowers, there is a larger barrier to overcome and the maximum lateral force before snapping will be greater. That is why the dissipation increases as the height lowers, as can be seen in both experimental and DFT-based simulations in Figure 2.

The new sentence now reads, “This is because the potential energy barrier that the surface presents to the apex becomes larger, as discussed later and shown in Figs. 4 e and f.”

The new figure 4 is

5. Figure 2b and 2d

The dissipation maxima for different bonds in Figure 2b do not agree with the simulations presented in Figure 2d. In Figure 2b, the dissipation maxima follow the order C-C(1) > C-C(4) > C-C(2) > C-C(3), whereas in the simulation shown in Figure 3, the order is C-C(1) > C-C(2) > C-C(4) > C-C(3). The authors should provide an explanation for this discrepancy.

(R1.14) Thank you for bringing up this point. We now directly address this in the main text. In the initial submission, we mentioned in the Methods that “While the experimentally-observed superstructure appears to be incommensurate, a commensurate structure was used as has been previously discussed in the literature⁴² as a necessary balance between accuracy and computational effort.”

We’re happy to bring this comment to the main text, and have added the sentences:

“We note that in the experiment, the largest dissipation is first found over C-C(1), then C-C(4), C-C(2) and finally C-C(3), whereas in the simulation, this order is C-C(1), then C-C(2), C-C(4) and then C-C(3). We tentatively attribute this to the fact that the simulation, by necessity, considers a two-molecule supercell, whereas the experiment probes PTCDA lying in an incommensurate lattice. At the height where we measure the PTCDA dissipation we do not see any impact from the copper substrate (discussed below), and therefore already minor changes in the bond length, such as those caused by the molecule-to-metal registry, can have a measurable impact.”

6. line 16 page 7

“We note that the DFT-based simulations do not include variations in the experimental oscillation amplitude and thermal effects..”

We understand that incorporating these parameters is challenging. The authors claim these effects lead to a smooth decrease in energy dissipation. We agree, yet if this is the case, thermal effects (and oscillation) should lower the peak height. However, the authors achieved excellent agreement between experiment and theory regarding the maximum values of E_{diss} , which seems odd.

(R1.15) An exploration of thermal effects for measurements that require picometer sensitivity of the dissipation channel is very challenging and currently outside the scope of this work. The data are all collected at liquid He temperatures, not room temperatures, so thermal fluctuations are much reduced from room temperature. Therefore we assume that the effects of thermal fluctuation at 5.8 K are not very large when compared to the maximum energy dissipation that is the topic of this manuscript. This is an ongoing research topic for us.

7. the last line on page 8.

“Consequently, this leads to greater energy dissipation.”

This is too simple. The authors should provide more explanation.

(R1.16) I hope that with the reply R1.13, in which we describe that what we are probing is related to a lateral energy barrier, that these sentences now make better sense.

8. Figure 3

The conclusion that energy dissipation correlates with bond order comes from Figure 3, which is based on simulations, not experimental data. As mentioned in comment 5, the relative magnitudes of the dissipation maxima obtained in experiments do not align with those from the simulations. So, the conclusion is not convincing. The experimental dissipation data corresponding to all C–C bonds should be provided. It is understandable that the orientations of different C–C bonds may vary, and molecule manipulation could help address this issue. Alternatively, studying a molecule with single, double, and triple bonds in the same structure would offer more convincing support.

(R1.17) There are two aspects we would like to address. First, in our response R1.14, we describe that we would expect small deviations anyway because the PTCDA form an incommensurate layer versus the simulated commensurate layer. As we mention in the main text and show in Extended Data Figure 1, the slopes are in excellent agreement.

Second, we also simulated the energy dissipation with an empirical potential. This we now mention in the main text:

“To verify the need for DFT-based calculations to determine the potential energy landscape, we also performed simulations using empirical atomic interactions^{24,33}. These results, shown in the Supplementary Notes and Supplementary Fig. 9, show a poorer agreement to the experimentally-determined E_{diss} .”

Therefore we assert that the DFT-based simulations, in which the electron density and is correctly described and the interaction of the apex CO molecule is better described (than the empirical model) are a good description of the tip-sample interaction.

Furthermore, we now provide experimental dissipation data of more bonds, shown in Supplementary Figure 7. We were not able to collect data over all bonds:

We were not able to manipulate PTCDA on Cu(111) with a CO-tip. We managed to pick it up with a metal tip but were not able to place it reliably on the surface. Most manipulation experiments of PTCDA are performed on Ag(111) where the interaction with the surface is weaker.

Finally, regarding the suggestion to examine a molecule with single vs double and triple bonds. We agree that this would be an excellent extension but is not the focus of the current work. Therefore we have carefully considered your feedback and decided to change the title as discussed in our reply R1.1.

9. It is recommended to also consider the C–O bonds within the molecule to determine if they follow the proposed correlation between bond order and energy dissipation.

(R1.18) This is an excellent recommendation. However, we found that collecting data over the C–O bonds of the PTCDA was very challenging. To understand that, we turned to the calculated electron density at various heights relative to the average height of the PTCDA above the surface. In the figure below, electron density is plotted at various heights above the molecule. As can be seen, at heights closer to our imaging conditions, the electron density is greater around the aromatic C–C bonds. This figure is now included as Supplementary Figure 4.

The PTCDA molecule bends so that the O atoms are lower than the C atoms (a point we discussed in Nam et al. PNAS 2024 – Ref. 23). This is also probably one reason that the potential energy landscape over the C–O covalent bonds was not conducive to E_{diss} measurements.

Considering O–C covalent bonds is an excellent idea for future studies.

10. Figure 4ef

It is difficult to understand the figures.

(R1.19) Thank you for bringing this to our attention. Please consider our reply R1.13 with the revised Figure 4. We hope that this display of the potential energy landscape is easier for readers to understand.

11. The discussion on dissipation over hydrogen bonds is insufficient. Why is the energy dissipation over some hydrogen bonds larger than that over covalent bonds? The order is a key factor for covalent bonds, what determines dissipation in the case of hydrogen bonds? Why different hydrogen bonds exhibit varying maximum dissipation values?

(R1.20) This is an excellent question. We included the observation of hydrogen bonds because of the surprising result (which we believe merits publication) that the maximum friction over hydrogen bonds can be as high as that over covalent bonds. Prompted by this reviewer's comment, we tried to determine a similar correlation for hydrogen bonds. However, there is not a monotonic relation between the maximum dissipation and the length of hydrogen bonds, which we now present in Supplementary Fig. 14:

And include in the main text:

“We were unable to find a similar correlation between friction and bond character for the OH bonds. In Supplementary Fig. 14 we show that the maximum energy dissipated over a OH bond is not a monotonic function of the distance between atoms. “

We propose that this is another observation that confirms that the mechanism for friction above OH bonds is fundamentally different than over CC bonds. This complex reason is, at its core, due to the potential energy landscape.

12. For such high-sensitivity measurements, has the author considered the effects of molecular vibrations. This point should be discussed.

(R1.21) The reviewer is correct that molecular vibrations are very important for several reasons, including that they allow us to consider the stiffness of the molecule in comparison with the soft CO at the apex. We have calculated these and show that they are much stiffer than the soft CO at the tip apex. We have included a section with our calculations in the Supporting Information: “Vibrational modes of PTCDA on Cu(111)”

The lateral spring constants of the vibrational modes are much higher than the lateral spring constant of the CO at the tip apex (0.24 N/m, as described in Ref. 22). A table from the SI, displaying the spring constants of the four softest modes, is reproduced here:

Vibration mode	Vibration frequency	Lateral spring constant
Frustrated translation	36 cm ⁻¹	31 N/m
Frustrated rotation	46 cm ⁻¹	50 N/m
Frustrated translation (2)	49 cm ⁻¹	55 N/m
Surface adsorbate	87 cm ⁻¹	175 N/m

The modes are identified by the corresponding Supplementary Fig. 15 in the SI.

Minor

1. Line 10 on page 3

“energy dissipated per cycle” of what?

(R1.22) Per oscillation cycle. The measurement is conducted with a tip that oscillates laterally. We have now changed the text to include this:

The average energy dissipated per oscillation cycle, E_{diss} , and the frequency shift, Δf (a measure of the conservative interaction), are simultaneously recorded.

2. Line 11 on page 3

“The heights of each curve (x-axis) were determined by the DFT...”

We suppose that the origin of the heights was determined by the DFT... or something missing?

(R1.23) This is correct.

Reviewer #2 (Remarks to the Author):

In this paper the authors propose that there is a correlation between the energy dissipation measured in frequency-modulation LFM experiments and the amount of electronic density in the region of “covalent” bonds estimated with Mulliken analysis from DFT calculations of the PTCDA adsorbed molecule. However, the correlation shown in Fig. 3, which should constitute the main result, is evaluated taking into consideration the simulated dissipation energy values, rather than the experimental ones. Indeed, the order of the Ediss maxima (c1, c4, c2, c3) resulting from the experiment, shown in Fig. 2b, is not the same as that (c1, c2, c4, c3) of the computational ones, shown in Fig. 2d. Moreover, in this latter case c4 and c2 are clearly different, while experimentally they are almost identical.

(R2.1) We agree with the reviewer. We now explicitly discuss this fact, as described in our reply R1.14. Please note that this has involved us making changes to the main text.

The computational Ediss strongly depends on the constraints imposed in the model, which in the present case are substantial, being frozen all the degrees of freedom of the tip, PTCDA, and substrate. With the CO the only unconstrained object in the model, it is hard to believe that the calculated Ediss can quantitatively resemble the experimental values. The failure to fully reproduce the experimental trend implies that important contributions to the dissipation are missing in the model. As a matter of fact the removal of almost all dynamical effects, with the force always calculated for the relaxed positions, appears to be questionable.

(R2.2) First, we would argue that using density functional theory to explore the potential energy landscape of the tip interacting with the surface is highly accurate and includes all relevant interactions including an accurate description of the electronic density of the substrate. However, we recognize that the reviewer is pushing us to explain the difference between the simulation and the experimental results.

As we mentioned in R1.14, we believe the major cause for the difference in maxima is that experimentally, PTCDA forms an incommensurate surface layer on Cu(111) and we are limited by computational power to explore a commensurate lattice with the DFT-based simulation. We have made changes (described in R1.14) to the main text.

We of course want to address any possible reasons for this difference and so we have also explored the vibrational modes of the PTCDA on the surface to see if there are missing dynamic effects. As we wrote reply R1.21, these modes have a much higher stiffness and therefore we expect that effects of the substrate’s degrees of freedom are small.

The correlation between dissipation and bond order is limited to four different C-C bonds, which makes it rather uncertain the generalization to any “covalent” bond as suggested by the manuscript title.

(R2.3) The correlation is proposed by the results of the simulation in which 15 aromatic bonds were probed.

The experimental data shown in Figure 2 are of four C-C bonds that have the same angle to the oscillation direction. We have acquired data over more bonds and include them in the SI, as discussed in our reply R1.17.

Furthermore, we agree with the assertion of the reviewer and have therefore changed the title as discussed in our reply R1.1.

Indeed, the fact that almost the same amount of dissipation is observed for the case of hydrogen bonds makes the connection to bond-order even less general and more specific to C-C bonds in the PTCDA molecule.

(R2.4) The interesting feature with dissipation over the hydrogen bond is that the sliding friction over it can be as big as over a covalent bond containing two C atoms. This might seem counterintuitive, as one might be tempted to label a hydrogen bond as a bond with order 0. To clarify this, we limit the discussion here to aromatic C-C bonds, as we now say in the title.

In order to explicitly address this, we have changed the main text to:

“Hydrogen bonds exhibit insignificant additional electron density between the bonding atoms (Supplementary Figs. 4 and 12). Therefore over hydrogen bonds, the dominant interaction is not with an increased electron density between the atoms (covalent bond) but rather with the atoms themselves³⁴.

The relationship that we observed between bond order and sliding friction over aromatic bonds does not hold for hydrogen bonds: The Milliken bond order of hydrogen bonds is small (Supplementary Fig. 12) and yet the magnitude of sliding friction is similar to that over aromatic bonds (Fig. 2). The reason that the dissipation is similar in magnitude is because the energy barrier presented laterally over hydrogen and covalent bonds are themselves of a similar shape, as shown in Figs. 4 e and f.”

In summary, although the paper presents very interesting experimental results their interpretation in relation to “sliding” friction is not convincing. The modeling of the process is not well explained even in the supplementary information.

(R2.5) The first reviewer had a similar comment, and we have made changes to the main text and major changes to the Supporting information as detailed in R1.8.

There are still unanswered open issues, such as the dissipation from the hydrogen bonds and the gaussian shape of the experimental Ediss as a function of the separation, and too many assumptions in the model and its interpretation, whose effects on the estimate of the dissipation remain unclear.

(R2.6) We have now addressed each of these points in R2.4 (H-bonds), R1.13 (the Gaussian shape), and R2.2 (the assumptions in the model).

Minor comments:

- Several important details of the studies have been confined to the Supplementary information, whose redaction has not been careful enough. Figures S2 show color coded surfaces, without any indication of which property is represented by the colors neither a colormap is shown to associate them to any actual value. Figure S2 is wrongly referenced several times (—> S3 ?)

(R2.7) Thank you for pointing this out. We have gone through the manuscript. All figures in the supplementary information are now mentioned explicitly either in the main text or Methods section.

- Definition of the color coding is also missing for figure 4e and 4f.

(R2.8) Thank you for pointing this out. As we wrote in R1.13, we have changed this figure, and the color is defined for Figs. 4e and f.

Reviewer #3 (copied from the attached pdf document):

This manuscript, titled "Sliding friction over individual covalent bonds correlates with bond order" by Nam et al., employs a highly sophisticated system to investigate the mechanisms of energy dissipation caused by C–C covalent bonds and hydrogen bonds during friction. Interestingly, seemingly identical C–C covalent bonds exhibit different levels of energy dissipation. Using DFT calculations, the authors determined the Mulliken bond order of the covalent bonds and found a linear relationship between bond order and energy dissipation, indicating that energy dissipation originates from variations in the electronic density of the covalent bonds. In the case of hydrogen bonds, the CO molecule does not interact directly with the hydrogen bond itself; instead, the energy dissipation primarily arises from the electronic density of the atoms involved in hydrogen bond formation. The experimental design of this study is highly innovative, and the observed correlation between energy dissipation and bond order undoubtedly provides deeper insights into the fundamental origins of friction. Nam and co-workers investigate the sliding friction at the atomic scale over individual covalent and hydrogen bonds using frequency-modulation lateral force microscopy (LFM) with a CO-functionalized tip. Their experiments reveal a wide variation in energy dissipation over chemically similar covalent bonds. Moreover, the maximum energy dissipation values over hydrogen bonds can be comparable to those over covalent bonds, and energy dissipation over hydrogen bonds occurs at a lower tip height than that over covalent bonds. The density functional theory (DFT) simulations reproduce the experimental data well. The wide variety in sliding friction over covalent bonds can be understood by the correlation between sliding friction and bond order, whereas for hydrogen bonds, the interaction is primarily with the atoms themselves rather than with a bond. The authors need deal with the comments before accepted by Nature communications.

(R3.1) We thank the reviewer for their review and have responded to each comment with changes both in the SI and main text.

1. In Figures 2b–e, how is the lateral axis value of tip height defined and determined?

(R3.2) The height of the tip is defined as the height of the O atom if it were unrelaxed. In other words, it is 300 pm closer to the surface than the height of the metal apex tip atom. These are then the heights over the average molecular height.

To clarify this, we re-phrased the sentence: "The heights of each curve (x-axis) were determined by the DFT-based simulation and represent the height of the unrelaxed O of the tip apex (300 pm closer than the metal tip apex atom) above the plane of the molecular adsorbates."

2. Within the range of tip heights used in the experiments, does the electronic density of the substrate atoms affect the energy dissipation of the CO molecule?

(R3.3) The average position of the PTCDA molecules above the Cu(111) surface atoms is 273 pm and therefore rather high above the underlying Cu atoms. We explored the influence of the underlying Cu substrate however both theoretically and experimentally.

In Supplementary Fig. 9 we compare the dissipation with and without a Cu substrate while freezing the geometry of the relaxed PTCDA adsorbates. The absence of the Cu affects the electron density of the PTCDA, which does have a small effect on the dissipation values (1-2 meV/cycle) that are quantitatively the same overall.

Experimentally, we have included a new Supplementary Fig. 8 to show that we do not observe dissipation over the bare Cu surface at heights where we see dissipation over the PTCDA island. To clarify this, we included a sentence in the Methods: Experimental section:

“Excitation at these heights was not influenced directly by the underlying Cu substrate, as can be seen in Supplementary Fig. 8. We also investigated the influence of the Cu substrate theoretically, as can be seen in Supplementary Fig. 9.”

And in the supporting information we include the following figure:

Supplementary Figure 5 | Excitation signal beyond the PTCDA islands. At heights where excitation can be observed over the PTCDA molecule, no excitation is observed beyond the edge of the molecular island. This is because the substrate does not directly contribute to the dissipation signal. The dark depression seen in the Δf image is a CO molecule and although this sits on the surface, there is no excitation signal observed over it.

3. Although the orientations of the C–C bonds at different locations are consistent, variations in the arrangement of substrate atoms beneath them may influence the measurement of energy dissipation. This is particularly critical for hydrogen bonds, where dissipation is measured at lower tip heights—care must be taken to exclude the influence of the substrate atoms.

(R3.4) This is again very correct. One of the defining characteristics of this dissipated energy is its very short decay length. That means that it is very sensitive to short-range interactions and that atoms further than a few tens of picometers away do not directly affect the measurements. To explicitly clarify this, we added the sentence:

“As we showed previously, the signal decays with a decay length of 4 pm, which is much smaller than those reported for STM or normal AFM measurements. It also means that the signal probes the potential energy landscape within a range of less than one Angstrom.”

4. The relative position between the tip and the covalent bond or atom directly affects the measurement. If the measurement point is too close to the atomic center, the atomic electronic density may dominate energy dissipation over the bond itself. How is the measurement position precisely controlled in such cases?

(R3.5) The measurement position can be controlled very precisely, as shown by the image in Supplementary Fig. 1. To highlight this, we include in the Methods: Experimental section the additional sentence

“Images of the area were then collected and linescans were taken between the atomic centres, as shown in Supplementary Fig. 1.”

5. Compared to the effect of covalent bonds on frictional energy dissipation, how significant is the contribution of individual atoms? In experiments using nanoscale or larger tips, where atomic electronic structures are more prominent at the surface, is the influence of covalent bonds significantly diminished?

(R3.6) When we initially observed this dissipation, the strong signal was above chemical bonds and not the individual atoms (consider, e.g. Fig. 1c), causing us to focus on them.

When we consider macroscopic friction, the typical picture we have is that of a nominally flat surface that ends in many small asperities. This is the first method that is proposed where the sliding friction of a single well-defined asperity can be measured. For a surface interface where the bond order can be tuned – even within a very small margin – we propose that this would have significant changes to the macroscopic friction. To make this clear, we have changed the Introduction (new text highlighted):

“Modern theoretical descriptions of friction have shown that **the main components to friction are small asperities**, and that a full understanding requires considering interactions over a range of length scales down to the atomic level¹⁸.”

And

“**While macroscopic friction is the product of many asperities interacting with a sliding surface, to address fundamental questions** it is essential to understand friction down to the level of single atoms^{12,14,19,20}.”

Thank you for this comment.

6. The manuscript refers to ‘Supplementary Information’ to explain additional details, but it would be helpful for readers if the authors could explicitly state which sections or figures in the Supplementary Materials correspond to specific statements in the main text. This would improve the clarity and accessibility of the work.

(R3.7) Thank you. As mentioned in reply R2.7, we have now explicitly mentioned each figure in the supplementary information in the main text or Methods section.

7. Please provide the characterization of the tip apex before and after the measurements to verify that the tip remained stable and that the CO molecule was not desorbed during the experiments.

(R3.8) These images, as well as a discussion, are now included in the supporting information under the Materials and Methods: Experimental section. The additional new text in the supporting information follows:

“CO-terminated tips are confirmed by imaging a CO on the surface. The tip termination is vital for reproducible images, therefore if the tip termination changes during scanning, the image contrast will change and it will be clear that the CO is no longer present. Nonetheless, we verified the presence of the CO before and after data collection, for instance the images of a surface CO with a CO on the tip before and after data of Figure 2:”

Supplementary Figure 2 | Characterization of the tip via a surface CO molecule. (a) CO imaged with a symmetric single-atom metal tip before CO pickup. (b) CO imaged with a CO-terminated tip prior to friction data acquisition. (c) CO imaged with the same CO-terminated tip after data acquisition, confirming stable tip termination throughout the measurement.

8. The authors need to further enrich the discussion and experimental evidence regarding hydrogen bonds. Currently, only two sets of hydrogen bond data are presented, which may not be sufficient to fully generalize the observed trends. Additionally, it would be valuable for the authors to discuss whether any structural or environmental differences exist between these two hydrogen bond configurations, and how such differences might contribute to the observed variations in energy dissipation.

(R3.9) Please consider our reply R2.4. In general, we attempted to determine an equivalent “rule of thumb” for hydrogen bonds but were unable. Please also note the new Supplementary Fig. 14.

9. If the Cu(111) substrate might influence the bond order of the PTCDA molecules and thus affect the measured energy dissipation, the authors should consider discussing this potential effect.

(R3.10) I hope that we have addressed this to your satisfaction in our replies R3.3 and R3.4.

10. Since friction at the atomic scale is known to exhibit anisotropy, it is recommended that the authors consider discussing the potential for anisotropy effects in their system, or propose future experiments to investigate this aspect, particularly for hydrogen bonds, which are strongly directional.

(R3.11) This is correct, and we have since made changes to the manuscript as described in our reply R1.12.

Reviewer #4 (Remarks to the Author):

We thank the reviewer for investing their time and effort in improving our manuscript.

Reviewer #1 (Remarks to the Author):

We (co-review with an early career researcher) appreciate the efforts that the authors have made to address our (concerns and those of both reviewers. We find that many of the issues have been resolved, and the manuscript has been improved.

Thanks very much for appreciating the improvements that have been made to the manuscript, and for your continuing work to improve our manuscript.

However, our main concern (comment 5) and Reviewer 2's main concern remain unaddressed. This point is central to the manuscript's main finding, as reflected in the title: "Sliding friction over individual aromatic bonds correlates with bond order." Although the analysis of the experimental data relies heavily on theoretical calculations, there is an inconsistency between experiment and theory. In the revised manuscript, the authors attribute this inconsistency to commensurate (theory) vs. incommensurate (experiment) lattices. While we understand that calculations for incommensurate lattices are nearly impossible, the underlying mechanism has not been clearly explained. The authors did not provide a clear rationale for why minor changes in bond length are caused by adsorption.

It is known (e.g. DOI: 10.1103/PhysRevLett.108.146103 and our previous work, shown in the Supplementary of Ref. 23) that the PTCDA strongly deforms from a planar configuration when adsorbed on Cu(111).

In order to relate different adsorption sites to different measured values of E_{diss} , we have performed further simulations. First, we used DFT-based methods to relax a pair of PTCDA molecules on the surface using two different unit cells to show the change in the molecular geometry upon adsorption. These changes are shown in a new Supplementary Figure 8.

Second, we show that different atomic positions of the PTCDA affect the measured dissipation signal. To do so, we compare calculated E_{diss} values between a planar gas-phase molecule and one in the adsorbed configuration, showing that geometric distortions have a clear effect on E_{diss} . These are explicitly shown in Supplementary Figure 9.

To incorporate these changes, we have changed the main text to:

At the height where we measure the PTCDA dissipation we do not see any impact from the copper substrate (discussed below), however there are geometric differences in different adsorption sites (Supplementary Fig. 8). These distortions from the gas-phase planar geometry affect the potential energy landscape and E_{diss} (Supplementary Fig. 9).

The authors also explained the mechanism of dissipation (R1.3). If this scenario is correct, one would expect to observe small lateral shifts in the peak position between forward and backward scans. The proposed mechanism also relies heavily on theoretical calculations, as there is no experimental data supporting this idea.

A small lateral shift implies that the fast and slow scan directions are relevant, i.e. that the starting point of the CO is highly relevant to the observed energy dissipation. However, we require repeated measurements of the snapping to be sensitive to it. We take into account the bandwidth, etc. and do not observe forward/backward asymmetry.

The mechanism which we propose has been supported using observations of energy dissipation at various amplitudes and heights which were all reported in Ref. 24 (our earlier work). That is to say that the mechanism is in agreement with all experimental data. In this manuscript, we go beyond our previous model to DFT-based modelling, improving the accuracy of the description of the tip-sample accuracy and show incredible agreement with magnitudes between the calculations and the experimental results. Therefore we argue that there is strong experimental data supporting our proposed mechanism.

The dissipation signal between the molecules, at the hydrogen bonding sites, remains unclear. The authors propose some complex possibilities, but is it possible that the molecule is simply being laterally moved by the tip's oscillation?

In order to consider the possibility of the molecules being deflected, we calculated the stiffnesses of their lowest modes on the surface, and reported them in the Supplementary Information section entitled "Vibrational modes of PTCDA on Cu(111)". This section includes the Supplementary Table 1, where the modes are shown to have a stiffness approximately 100 times higher than the CO at the apex. Therefore we do not think the molecule is being moved by this soft tip apex. To clarify this point, we re-phrased the sentences in the Methods section (under DFT modelling) to emphasize this:

The low-energy vibrational modes of the PTCDA adlayer were also investigated (Supplementary Fig. 17 and Supplementary Table 1). The corresponding stiffnesses of these modes are approximately 100 times higher than that of the CO, indicating that relaxation of the molecule can be ignored during the snapping.

Regarding R1.15, "Therefore we assume that the effects of thermal fluctuation at 5.8 K are not very large." I fully agree with this statement, but in the main text the authors write, "We note that the DFT-based simulations do not include variations in the experimental oscillation amplitude and thermal effects which we believe are responsible for the smooth decrease in energy dissipation at heights below the

maximum.” We think the phrase “we believe are responsible for the smooth decrease in energy dissipation at heights below the maximum” conflicts with the authors’ previous argument.

The effect of temperature with these measurements is very important, especially as it would allow even greater agreement with theory. We have in the meantime investigated this issue which we intend to publish in the future. We will not discuss these results in this reply to be published, but are happy to privately share with the reviewer.

We fully agree with Reviewer 2’s comment: “Their interpretation in relation to ‘sliding friction’ is not convincing.” The definition of “sliding friction” is unclear, even though the authors replied to our comments (R1.8). Thus, the current title might not be appropriate.

Sliding friction is a non-conservative force that acts when two objects slide against each other (are relative to each other in motion). Non-conservative forces are characterized as those that do work when an object is moved in a closed path. In this case, the metal apex slides over the surface and work is done on the sensor which is recorded as the energy dissipation. For clarity, we have included the following sentences in the Discussion section:

Sliding friction is a non-conservative force that acts when two objects slide against each other and opposes the relative motion between the two surfaces. Non-conservative forces are those that yield non-zero work when an object is moved in a closed path. During each oscillation, the metal apex is slid forward and backward over the surface in a closed path, and if work is done on the sensor it is recorded as the energy dissipation.

Minor:

The term “torsional spring” sounds odd, as there is no torsional movement of either the tip or the molecule.

We use this term because the energy stored is a function of the deflection angle. However, we believe that the term more familiar to chemists is “angle bending”. We have therefore added the following sentence to the main text:

This deflection is also referred to as angle bending.

This manuscript, titled "Sliding friction over individual covalent bonds correlates with bond order" by Nam et al., employs a highly sophisticated system to investigate the mechanisms of energy dissipation caused by C–C covalent bonds and hydrogen bonds during friction. Interestingly, seemingly identical C–C covalent bonds exhibit different levels of energy dissipation. Using DFT calculations, the authors determined the Mulliken bond order of the covalent bonds and found a linear relationship between bond order and energy dissipation, indicating that energy dissipation originates from variations in the electronic density of the covalent bonds. In the case of hydrogen bonds, the CO molecule does not interact directly with the hydrogen bond itself; instead, the energy dissipation primarily arises from the electronic density of the atoms involved in hydrogen bond formation. The experimental design of this study is highly innovative, and the observed correlation between energy dissipation and bond order undoubtedly provides deeper insights into the fundamental origins of friction. Nam and co-workers investigate the sliding friction at the atomic scale over individual covalent and hydrogen bonds using frequency-modulation lateral force microscopy (LFM) with a CO-functionalized tip. Their experiments reveal a wide variation in energy dissipation over chemically similar covalent bonds. Moreover, the maximum energy dissipation values over hydrogen bonds can be comparable to those over covalent bonds, and energy dissipation over hydrogen bonds occurs at a lower tip height than that over covalent bonds. The density functional theory (DFT) simulations reproduce the experimental data well. The wide variety in sliding friction over covalent bonds can be understood by the correlation between sliding friction and bond order, whereas for hydrogen bonds, the interaction is primarily with the atoms themselves rather than with a bond. The authors need deal with the comments before accepted by Nature communications.

1. In Figures 2b–e, how is the lateral axis value of tip height defined and determined?
2. Within the range of tip heights used in the experiments, does the electronic density of the substrate atoms affect the energy dissipation of the CO molecule?
3. Although the orientations of the C–C bonds at different locations are consistent, variations in the arrangement of substrate atoms beneath them may influence the measurement of energy dissipation. This is particularly critical for hydrogen bonds, where dissipation is measured at lower tip heights—care must be taken to exclude the influence of the substrate atoms.
4. The relative position between the tip and the covalent bond or atom directly affects the measurement. If the measurement point is too close to the atomic center, the atomic electronic density may dominate energy dissipation over the bond itself. How is the measurement position precisely controlled in such cases?
5. Compared to the effect of covalent bonds on frictional energy dissipation, how

significant is the contribution of individual atoms? In experiments using nanoscale or larger tips, where atomic electronic structures are more prominent at the surface, is the influence of covalent bonds significantly diminished?

6. The manuscript refers to 'Supplementary Information' to explain additional details, but it would be helpful for readers if the authors could explicitly state which sections or figures in the Supplementary Materials correspond to specific statements in the main text. This would improve the clarity and accessibility of the work.

7. Please provide the characterization of the tip apex before and after the measurements to verify that the tip remained stable and that the CO molecule was not desorbed during the experiments.

8. The authors need to further enrich the discussion and experimental evidence regarding hydrogen bonds. Currently, only two sets of hydrogen bond data are presented, which may not be sufficient to fully generalize the observed trends. Additionally, it would be valuable for the authors to discuss whether any structural or environmental differences exist between these two hydrogen bond configurations, and how such differences might contribute to the observed variations in energy dissipation.

9. If the Cu(111) substrate might influence the bond order of the PTCDA molecules and thus affect the measured energy dissipation, the authors should consider discussing this potential effect.

10. Since friction at the atomic scale is known to exhibit anisotropy, it is recommended that the authors consider discussing the potential for anisotropy effects in their system, or propose future experiments to investigate this aspect, particularly for hydrogen bonds, which are strongly directional.